# From Rational Inquiry to Sacred Insight: The Role of Religion in Augustine's Views on Liberal Education

Jeong-In Lee [1] and Jangwan Ko [2,*]

1    Institute for Poverty Alleviation and International Development, Yonsei University,
     Wonju 26493, Republic of Korea; epimess@yonsei.ac.kr
2    College of Education, Sungkyunkwan University, Seoul 03063, Republic of Korea
*    Correspondence: jakosu@skku.edu

**Abstract:** This paper examines the role of religion in liberal education based on the Christian thinker St. Augustine. In his early work, *On Order*, Augustine posited that through rational inquiry, as epitomized by rational knowledge learned by the *trivium* and the *quadrivium*, one can understand the order of the world and eventually obtain divine truth. However, in *On True Religion*, he withdraws from this position and instead emphasizes that rational knowledge has three limitations: First, regarding the foundation of knowledge, rational knowledge can inform about what things are, but it fails to explain why things exist in the manner they do. Second, concerning the purpose of knowledge, rational knowledge can elucidate the attributes of things, but it falls short in providing the ultimate goals to which these things aspire. Third, concerning the acquisition of knowledge, rational knowledge seeks extroverted knowledge, i.e., knowledge that is oriented toward external objects without introspecting on the inner self. In light of Augustine's emphasis on the limitations of rational knowledge, the current study provides two possible interpretations of the relationship between liberal education and religion. One is an active interpretation which posits that, by resolving its limitations, religion can fully replace liberal education. The other is a passive interpretation, which suggests that religion can illuminate the boundaries of liberal education and refresh them, thereby enabling the learner to deeply reflect on knowledge and connect it with their inner self.

**Keywords:** Augustine; liberal education; religious education; *On True Religion*; rational knowledge





## 1. Introduction

Augustine's religious ideas have exerted substantial influence on not only philosophy (Copleston 1990; Plantinga 1992) but also education (Marrou 1948; Rappe 2001). St. Augustine has particularly emerged as a key figure who laid the groundwork for systematizing liberal education. Liberal education, which was considered an enduring educational ideal even in modern times, was first organized in the Middle Ages with the introduction of the seven liberal arts (*Septem artes liberales*) structured in a systematic curriculum. In contrast to Plato and Aristotle, Augustine played a crucial role in interpreting the liberal arts in religious ways, specifically by linking them to the Christian God. St. Augustine's religious perspective on liberal education has been discussed academically (Kenyon 2012; Paffenroth and Hughes 2000; Pollmann and Vessey 2005), and it is often specifically considered in the construction of liberal arts curricula in universities (Chiariello 2015; Scott 2015; Whitfield 2015).

Among the *Cassiciacum dialogues* that represent his early ideas, *On Order* (*De ordine*) is a key text in which the young Augustine presented his theory of liberal education (St. Augustine 2020). In this book, God is regarded as a Philosophical God, in relation to whom Augustine believed that a few outstanding people with "well-trained souls" could reach the divine truth (Brown [1967] 2000, pp. 114–16). He had the ambition of reimagining the concept of liberal education not as a discipline that would enable the seeking of secular

happiness in the existing world, but as a religious practice that would enable the seeking of higher knowledge, of understanding and loving God, and finally of attaining happiness (*beatitudo*) (*ord*. 2.9.26).[1] Augustine then began to write specific treatises on the seven liberal arts *(retra*. 6.1). He completed *On Grammar* (*De Grammatica*), wrote the main part of *On Music* (*De Muscia*) (Aquila 2022), and drafted the other treatises on *Dialectics, Rhetoric, Geometry, Arithmetic*, and *Philosophy*.

However, Augustine put this project on hold after his Christian baptism in 386, and he finally abandoned it in 395. In his later years, he found that he had lost all but a few pages of *On Music*, but he did not regret this loss (*retra*. 5.3). This suggests that Augustine's interest in liberal education had waned and that he had begun to regard liberal education rather critically (Brown [1967] 2000, pp. 261–62; O'Meara 2001, pp. 155–60; Topping 2010, pp. 75–76). This change in attitude can be seen in *Retractations* (*Retracatione*), a review he undertook in his later years of all his earlier writings. Reviewing *On Order*, he withdraws his emphasis on liberal education in the book, observing:

> "Many saintly persons do not know much—some, in truth, know them and are not saintly . . ." (*retra*. 3.2).

In *On Order*, written by the young Augustine prior to being baptized, learning liberal arts is posited to be a primary avenue to understanding God's truth. By contrast, the old Augustine, as a bishop, does not believe that learning liberal arts guarantees that one will become a saint. The current paper attributes this shift to Augustine's reconsideration of the value of religion after his conversion to Christianity. In other words, he later discovered a "deeper" dimension to religious values, which led to his decision to be baptized and later his retraction of his optimistic view of liberal arts. Moreover, in his *Confessions*, Augustine describes the grammar and rhetoric he learned in his childhood as nothing more than "linguistic skills" (*linguosae artes*, *conf*. 1.9.14) and "vanity" (*vanitas*, *conf*. 1.18.28), implying that such liberal education was futile as it could not prevent him from a life of lust (*conf*. 4.16.30). These expressions indicate that his view of liberal education had become more critical or pessimistic (Paffenroth and Hughes 2000, pp. 96–98; Burton 2007, pp. 65–81).

Given that this change in attitude can be observed after his baptism, it is natural to assume that having a more profound understanding of religion influenced his perspective on liberal education (O'Donnell 1992a, p. xlii). The current paper then raises the following questions: What are the values of religion that he thinks are lacking, or are inferior, in liberal education? What does he consider to be the ideal relationship between liberal education and religion?

The purpose of this study is to examine the role of religion in liberal education according to St. Augustine. Previous studies have focused on *On Order* in discussions with liberal arts education (Kenyon 2012; Paffenroth and Hughes 2000). However, because these studies only deal with Augustine's early thoughts, they have the limitation of ignoring Augustine's later thoughts when his stance on liberal education changed. This study examines Augustine's transformation from a rational view of liberal education to a religious view, and examines Augustine's later thoughts to comprehensively understand Augustine's views on liberal education. To this end, we first pay attention to the changing point where his religious view overtakes his rational view in education, and we then examine *On True Religion* (*De vera religione*), which is the first religious text Augustine wrote after he ceased writing about liberal education. Finally, we suggest two possible relationships between reason and religion in Augustine's thought from a contemporary perspective.

## 2. Liberal Education in the Roman Era

The origins of liberal education can be traced back to ancient Greek philosophy. The spirit of liberal education and discussions on liberal education that were initially advocated for by Plato were further specified by Aristotle, and the need for such liberal education was passed down to greater Roman society (Brubacher 1966, pp. 243–57; Marrou 1948, pp. 61–78, 242–46). However, it was not until the Middle Ages that the seven liberal arts in the typical sense of liberal education were established (Kimball 1995); Augustine

contributed greatly to the formation and development of liberal education during this period. In other words, the tradition of liberal arts, which originated in Greece, firmly took root in medieval Christian society through Augustine's contributions.

As Christianity became the state religion during the Roman era, liberal education emerged as a very important issue for Christians. While some Greek intellectuals such as Plato argued that humans could attain the highest knowledge and ascent (*anabasis*) through the pursuit of a rational mind, acquired by inference and contemplation, Christianity instead maintained that the highest knowledge could only be achieved through the religious pursuit of God (Joo 1998). This new perspective exerted a particularly large influence on education curricula. At the time, the church considered Greek liberal education subjects—such as grammar and rhetoric—to belong to pagan culture, so whether and how the church would accept such education represented a major issue. At this time, Augustine held the position that it was necessary to borrow from pagan culture, while maintaining that the acceptance of liberal arts did not go against the Christian faith. He acknowledged that liberal education has the common educational purpose of making learners free, specifically by leading them from the level of sensory experience to the level of pure intellectual inquiry (Howie 1969).

Augustine emphasized the order of things as a framework that allows us to see reality and as an avenue allowing for the pursuit of true happiness in the world. The order is both a principle that controls individual things and a principle that applies to all things, so it has both particularity and universality (*ord.* 1.1.1). In this respect, knowing the order of things means knowing the order of the world to which things belong; further, in that God created the world, knowing this order is also to know God's order.

Augustine argues that specific studies and procedures must be followed to understand God's order in all things. This is what we refer to in today's subjects of liberal arts as the *trivium* (grammar, rhetoric, dialects) and *quadrivium* (arithmetic, geometry, music, astronomy) (*ord.* 2.2.7). Although the archetypes of these subjects were first introduced in Plato's *Republic* (*Politeia*, 525b–533e) (Plato 2004), Augustine highlights a religious sense within them. By understanding these seven liberal arts subjects, we acquire the ultimate knowledge with which to understand the divine order and reach the highest principles it contains. We can refer to this activity of understanding and reaching as 'doing philosophy' *(philosophia)*, which leads us to the highest principle, i.e., the principle of oneness or the One *(to hen)* (*ord.* 2.18.48).

However, Augustine's initially positive attitude became more negative in his later works. In *On Christian Teaching* (*De doctrina Christiana*), Augustine distances himself from the writing of liberal education. In that book, he offers a warning to readers who want to learn rhetorical skills (*doctr.chr.* 4.1.1.), he underscores the use of rhetoric in liberal arts (*doctr.chr.* 2.13.20, 2.36.54), and he emphasizes the superiority of Christian culture over pagan culture, including liberal education (*doctr.chr.* 2.38.58, 2.41.62–63).

### 3. An Intersection of Liberal Education and Religion: Augustine's *On True Religion*

The first text in which Augustine presents his different attitude to liberal education is *On True Religion (De vera religione)* (O'Donnell 1992b, p. 27). Written in 390, *On True Religion* was the last text Augustine wrote before being ordained a priest in the city of Hippo, and it is regarded as both a summary of his philosophical journey up to that point as well as a summary of the religious doctrine that would shape his remaining years (Fleteren 1999, p. 864). Thus, though *On True Religion* has not received much scholastic attention compared to his prior works, it is a crucial clue to understanding why Augustine criticized the liberal education that he once favored.

As the title suggests, *On True Religion* is a text in which Augustine aims to identify the true religion and its features. It mainly discusses theological subjects such as refutations to Manicheanism, an ascent to God, the Trinity, and the problem of body and soul (Fleteren 1999, pp. 476–77; Cary 2000, p. 35; Fox 2015, chp. 29). However, if we focus on when this text was written, *On True Religion* shows his earlier thoughts denying that liberal education is a form of learning in the pursuit of truth. Although Roman polytheism and Manichaeism

are typical targets of criticism that are positioned as being in contrast to the "true religion", some disciplines called "philosophy" that contrasted with the Christian doctrine at the time are also discussed critically (*ver. rel.* 1.1).[2] Indeed, Augustine explains, one of the purposes of the book is to discuss religion and philosophy together, for the benefit of "those who do not carry philosophy into religious observance or philosophize in a religious spirit" (*ver. rel.* 7.12). Given that philosophy is considered to be the final form of liberal education, *On True Religion* can be regarded as Augustine's attempt to relocate the relative positions of liberal education and religion.

Beginning with *On True Religion*, Augustine no longer takes the optimistic view of human reason that had characterized his earlier educational thought. This signifies his departure from the Platonic notion that divine truth can be attained through the practice of rational activities in liberal arts (Dobell 2009, p. 203). In *On Order*, Augustine states that a few talented philosophers could reach true happiness by attaining the divine truth in liberal arts (*ord.* 2.19.47), whereas in *On True Religion*, he emphasizes that a heavenly life with knowledge is impossible for anyone in this life to achieve, except through divine grace (*ver. rel.* 12.24). Here, Augustine contrasts philosophy and religion, privileging the latter. What led him to view philosophy as having a limited ability to attain the truth? Augustine's writings suggest that there are two routes to an answer to this question: One is to criticize the contents of philosophy in opposition to the doctrines and spiritual events in the Bible (*ver. rel.* 3.4, 8.14), while the other is to reveal the weaknesses and limitations of philosophy in light of the roles of religion. In the present work, we explore the latter approach, as it is more relevant to the topic of this study. By highlighting the limitations of rational knowledge, this approach contributes to the appreciation of the religious dimension of liberal education without necessarily embracing specific religious creeds. Taking this approach, the following chapter lists relevant passages from Augustine's book, *On Order*, that focus on criticizing rational knowledge, i.e., the knowledge produced by human reason, as represented by liberal education.

## 4. Limits of Rational Knowledge

In *On True Religion*, as well as in the earlier *Cassiciacum* writings including *On Order*, Augustine emphasizes the importance of human reason. By engaging in rational activities, we can perceive what is immutable among sensory and changeable things, which allows us to grasp the principles behind visible things (*ver. rel.* 29.52). This rational knowledge enables us to see things harmoniously arranged as a whole (*ver. rel.* 40.76). However, whereas *On Order* emphasizes the pursuit of truth through rational activities, which is similar to Platonic philosophy, *On True Religion* defends the superiority of Christianity over Platonic philosophy at various points (Fox 2015, pp. 397–402). In this respect, in *On Order*, Augustine recognizes the value of rational knowledge while also stressing its limitations.

In the following, we illustrate three aspects of Augustine's critique of rational knowledge in *On True Religion*. These aspects are related to the foundation, purpose, and acquisition of rational knowledge. In this approach, we examine the status of religion in Augustine's thought and discuss these aspects from a contemporary perspective.

### 4.1. The Foundation of Knowledge

Augustine does not deny the reliability of rational knowledge. While the knowledge from what others say or certain authority is at most probable (*probabile*), rational knowledge is certain (*certus*) in that we accept this knowledge under a process of understanding (*ver. rel.* 8.14). With inference and interpretation, we ascertain which knowledge is more certain than the other one. We also feel a sense of beauty and pleasure when perceiving things in harmony and balance. However, in *On True Religion*, Augustine raises further questions: What makes us consider some knowledge as certain or enjoyable? By what criteria do we judge things? Augustine believes these questions to be ultimately related to liberal arts because arts (*artes*) are sets of the most advanced knowledge we perceive as being important and worthwhile in relations to rational activities. Why does *reason* attribute this

sort of art as being important? In approaching this question, Augustine illuminates the ground of rational knowledge that we take for granted.

> "Now we must ask what is the nature of an art. By an art in this context I would have you understand not something that is observed by experience but something that is found out by reason. There is nothing very remarkable in knowing that sand and lime bind stones more securely together than mud, or that he who would build elegantly, must put a feature that is to be unique in the middle of the building, and if there are several features, they must be made to correspond, like with like. That is sensuous knowledge, but it is not far from reason and truth. We must indeed inquire what is the cause of our being dissatisfied if two windows are placed not one above the other, but side by side, and one of them is greater or lesser than the other, for they ought to have been equal" (*ver. rel.* 30.54).

Here, Augustine examines the fundamental nature of rational knowledge, taking arts as an example. We attain different types of knowledge from various situations. One type is knowledge through experience, while the other is knowledge through reason. An example of the former is that 'sand and lime have a high adhesive power when mixed together', while an example of the latter is that, when the parts of a building have symmetry and unity, we consider it to be satisfying. This sense of satisfaction is obtained through the senses, but it also has universal truth which can be applied to all situations. We may classify these two kinds of knowledge according to Plato's dualism into sensory and rational knowledge, and then give priority to the latter. However, Augustine goes one step further, indicating that what is important is that the latter is accepted as self-evident, even though there is no empirical basis to think so. The pleasures of symmetry and harmony are felt and known through reason directly and intuitively. So, he asks: Why do we feel satisfied with symmetries and harmonies, and why do we feel uncomfortable with the opposite? What is the cause of this? What are the reasons? On what *basis* do we judge being symmetrical and harmonious to be beautiful? Augustine points out that reasoning may guarantee that the knowledge it produces is certain, but it does not guarantee that it has values such as importance or beauty. However, our mode of thought includes this kind of judgment (*iudicare*) on the results of reasoning (*ver. rel.* 30.54).

> "What is it in us that enables us to judge all these, the plan they are following and how far they accomplish it; to judge ourselves, too, in our buildings and other activities of the body, as if we were lords of all such things?" (*ver. rel.* 43.80).

Even if we know that what pleases us is balance and harmony, it is another matter to know *why* we are pleased by it, or why we consider things to be beautiful and perfect (*ver. rel.* 30.55). Similarly, while we may know the truth of a thing through reasoning, we may not know why it is true, or why we regard it as being true. We may calculate that a star travels in a certain cycle, determine that a substance has certain properties, and find an image beautiful, but we may not know why the star travels in a certain cycle, why the substance has certain properties, or why we find the image beautiful. "No one can say why these intelligible things are as they are." (*ver. rel.* 31.57).

However, we do naturally make certain judgments about things when they show certain properties or arrangements. We take such things for granted and do not raise questions regarding such certainties. Here, Augustine points out the limitation of reasoning in that it cannot provide an answer to the foundation it forms. He argued that the judgments we naturally come to do not come from reasoning, but may be shown by reasoning; that is, they are universal truths that have already been rooted in our mind prior to the reasoning process. The mind has some sort of standard by which it makes judgments about things, but the mind does not make judgments about that standard itself, so these standards are not revealed by rational knowledge. "This standard of all the arts... the standard which is called truth is higher than our minds." (*ver. rel.* 30.56). Human reason takes the norms, forms, and examples of truth for granted, and knowledge is based on them (*ver. rel.* 31.58).

We use rational knowledge, but we do not inquire into its grounds, taking it for granted as natural, or we may not even be conscious of it. According to Augustine, these grounds are divinely made and related to God, and we accept these as true, though we do not understand them (*ver. rel.* 31.57). Human reason deduces and judges the given, not showing its nature; although we may understand knowledge, we may not explain why it exists, or how it came to be (*ver. rel.* 31.57–58).

Rational knowledge is limited in that it does not know its own foundations. Although we may express and articulate our minds through language, and although we may know the principles of numbers that govern the universe through astronomy, we may not be able to answer how it is possible to express our minds through language, or why numbers should be a principle of the universe. In this respect, knowledge through reason is limited in that it cannot know everything; it is a type of knowledge that is missing a key point, particularly in that it cannot know the basis for the existence of important things in our lives.

### 4.2. The Purpose of Knowledge

Reason enables us to perceive and judge things, but it does not know why these perceptions and judgments are true. This ignorance relates to not only how things are made, but also what they are for, i.e., their purpose. Reason may be able to give certain answers to the question of how we exist (*qui*), but it cannot answer the question of why we exist (*cur*) (*ver. rel.* 31.56). The question 'why' is concerned with the origin of things, but it is also concerned with the purpose of things. This teleological view, which was initially triggered by Aristotle, dominated Medieval thought, and Christianity further developed according to this perspective (Rocca 2017; McDonough 2020). However, even without scrutinizing whether the teleological view is true or false in explaining the world, we may consider that having some purpose is an important aspect of leading a meaningful life.

Similarly, for Augustine, asking why our knowledge is established is a question that is not only about the foundations of our knowledge but also about its purpose, particularly its *raison d'être*. For example, a person can be understood through reason by investigating empirical data, such as their gender or race, the places and situations they have lived in and experienced, and their character and tendencies. However, reason cannot give definite answers to questions such as what human beings strive for, what is meaningful to human beings, and what must be known to know it.

Augustine thinks that these sort of 'why' questions about the purpose of things can only be answered by religion. The rational man discovers knowledge and uses it; however, the religious man goes further and asks what this knowledge is finally used for (*ver. rel.* 46.87). Here, the final use is contrasted with the immediate use of things. For example, a doctor's medical knowledge may provide them with a high and stable income, while also providing them with the self-esteem that comes from saving human lives. The question of 'purpose' may not aim for visible outcomes, but it may instead have a lot to do with the nature of being human, or the meaning of existence. Augustine believes that human beings can have knowledge of facts, skills, and even principles, but not knowledge of the meaning of their own existence. Scott-Craig distinguishes the former as "knowledge about things" and the latter as "knowledge of things". Knowledge about things is the knowledge of some properties of things, while knowledge of things involves the knowledge of the *raison d'être* of things. Moreover, while 'knowledge about things' can be grasped by the human intellect, 'knowledge of things' necessarily has a religious dimension that belongs to the realm of God and is thus unattainable by the self, except by divine grace (Scott-Craig 1979, pp. 128–29).

Augustine believes that human beings are superior to other creatures because they search for the meaning of existence, and because they can recognize themselves anew in doing so (*ver. rel.* 44.82). Reason cannot elucidate what we live for and why we sustain our life or which life is significant for us. The understanding of purpose can only be achieved by seeing the whole, reaching from beginning to end, while the faculty of human beings is limited for this view. Therefore, although human reason can understand the present

state of things, it does not know what they will eventually mean, or for what purpose they were given.

*4.3. The Acqusition of Knowledge*

Given that acquiring and understanding knowledge is a matter of the soul or mind (*animus*), it seems obvious that a comprehensive understanding of the nature of the mind is a crucial prerequisite for the adept attainment of knowledge. In a religious sense, the connection between knowledge and the inner self is related to the way knowledge is acquired in the mind. Liberal education, and that teaching the knowledge associated with the *quadrivium* in particular, seeks external knowledge in the world, such as the mathematical principles behind physical circumstances. Liberal education is a rational activity whose basic role is to express the mind outwardly in the form of the *trivium* and explore principles that are outside of the mind as the *quadrivium*. This outward pursuit of knowledge, which 'seeks truth from the external', may lead an individual to think that the pursuit of knowledge fundamentally exists apart from the one's mind and that the pursuit of knowledge is merely an extension of curiosity (*ver. rel.* 49.94). In this case, knowledge is external in both senses, i.e., as a way of attaining information and as a way of achieving separation from the individual's mind.

Religion, on the other hand, seeks knowledge within. To seek knowledge within means to look inward and find truth within oneself. The main concern of religion is not the rational soul, but the inner truth that establishes our existence (*ver. rel.* 55.110). The pursuit of knowledge is an activity of the mind, and reason operates with one's mind as its cause along with one's interiority as a background. Therefore, the mind is superior to any other route to the truth. However, this superiority is not due to human beings, but to the God within one. Ascension of the soul is therefore possible because there is a trace of God within one and all ([Fox 2015](), pp. 401–2).

If we focus on interiority, it is not possible for being and knowledge to be separable realms. While liberal education has limitations in its ability to achieve understanding of an object, inner knowledge through religion connects the 'I' with the knowledge, thus providing the reason for and meaning of our existence. Reason, though a great faculty, is not sufficient to understand something such as that needed to live a good life; even if one can attain lofty truths, his soul is easily corrupted without the will and practice to follow them. In religion, the important thing is not attaining knowledge but living in accordance with the truth; likewise, it is more important to follow and imitate God than it is to attain the knowledge of God (*ver. rel.* 47.90). Even if one is to master a liberal education and accomplish outstanding outcomes with the knowledge they have obtained therein, Augustine maintains that one cannot live a flourishing life without a map giving them a purpose and reason.

## 5. The Place of Religion against Liberal Education

Reason is a key faculty of human beings that helps us understand the order of the world based on the principles and features of things. However, Augustine demonstrates that reason is not able to provide every type of knowledge, and that it has limitations in understanding the divine order, which is the ultimate grounds and purpose of existence. As discussed above, rational knowledge has three limitations: first, although reason may help identify features or properties of things, it does not provide the answer to why things exist in any given manner. Second, reason cannot suggest the purpose of things that constitute a meaningful life, particularly for human beings. Third, the outward nature of rational activity often leads us to overlook inward exploration and therefore fails to see the fundamental relationship between knowledge and mind. Augustine continued emphasizing the limitations of knowledge in his later works in ways that evolved with his maturing theology (*doctr.chr. 2.38.57; trin. 9.6.10, 12.14.23 14.5.7; civ.dei. 11.28, 22.29*)

For Augustine, the limits of rational knowledge are synonymous with the limits of liberal education. He maintains that liberal education can produce some useful results

through rational activities, but that it cannot show the grounds on which it stands, the ends to which it is directed, and the connection between liberal education and the existence of the self. Religion proclaims that it is the divine order that established liberal education, that it is the truth of God that should be pursued by liberal education, and that knowledge and existence are connected within God. While liberal education plays a limited role in grasping the truth of things, religion provides essential constituents of knowledge by giving purpose and meaning to things. In other words, religion enables us to see things with a new perspective, particularly in terms of truth and its relevance to the self:

> "One God alone I worship, the sole principle of all things, and his Wisdom who makes every wise soul wise, and his Gift whereby all the blessed are blessed. . . . What good friend will the man lack who worships the one God whom all the good love, in knowing whom they rejoice, and by having recourse to whom as their first principle they derive their goodness?" (*ver. rel.* 55.112).

Understanding the grounds, purpose, and meaning of knowledge is suggested to constitute a way to true happiness. This happiness is not attained by acquiring something external but is instead achieved in a process of inward growth, a renewal of knowledge in the self (*ver. rel.* 26.48). What makes this inner renewal is not the rational mind, but the divine "illumination" (*illuminatio*) of the true meaning of things that are already known (*ver. rel.* 28.51).

Given that, for Augustine, the knowledge gained through liberal education has obvious limitations, then should liberal education be discarded as no longer valuable? Although Augustine does not explicitly answer this question, he reexamines the significance of liberal arts from the perspective of biblical studies in his later work, *On Christian Doctrine*. However, controversy persists as to whether this reassessment implies Augustine's promotion of a restricted application of liberal education or its complete abandonment (Topping 2010, pp. 75–78).

In the contemporary context, liberal education has undergone a significant transformation, shifting from being a representative form of comprehensive education to serving as a preparatory foundation for more specialized studies. The question of the relationship between liberal education and religion revolves around the interpretations of the respective aims and roles of each. Whereas liberal education, represented by language, mathematics, and science, aims to find epistemological external truth through the use of reason, religion seeks ontological and internal truth, emphasizing the inner self. Given this difference, there are two possibilities for interpreting the position of religion in Augustine's theory of liberal education: one is an active interpretation whereas the other is a passive interpretation.

The active interpretation suggests that the general role of reason in liberal education would yield to the influence of faith and intuition in religion. This interpretation would suggest that the content of liberal education should be restructured in consideration of religious foundation and purpose, thereby evolving into a form of religious liberal education. However, this interpretation ultimately necessitates the acceptance of specific religious doctrines, such as monotheistic concepts and the rituals associated with a particular faith. This raises the question of how favorable this proposition is for modern students who navigate their studies and lives without any religious affiliation. Even if we overlook this unfamiliarity, it appears to be a logical leap to propose that recognizing the limitations of liberal education should entail embracing the particular doctrines and practices of religion.

An alternative interpretation is the passive interpretation, which suggests that the differences between rational knowledge and religious knowledge can be complementary. In this view, both reason and faith play a role in shaping knowledge under the same goal of seeking truth. On the one hand, religion highlights the limits of rational knowledge while warning of the arrogance and self-righteousness that can accompany reason. On the other hand, rational knowledge can provide rationales and justification for the blindness to which religion can be prone. This mutually beneficial relationship can lead to each perspective illuminating what the other has failed to see and ultimately creating a holistic

understanding of knowledge. We believe that this passive interpretation is more apt for modern society and provides more productive implications in education.

This complementary view of liberal education and religion, while not endorsing the doctrines of any particular religion, suggests a new dimension of liberal education. Thinking of knowledge as connected to the inner self can have a different resonance to what and how we offer education. If we are conscious of the fact that knowledge is used to elucidate the existence of the 'I', then we can reach a new dimension of understanding, even when considering the same objects as before. For example, when we ascertain that the inherent value within our inner being is that associated with leading our life, we experience a sense of self-respect or reverence for our existence in the human condition. This transcends the mere acquisition or understanding of truth; it is a path that leads us to love truth and live in accordance with it (*ver. rel.* 47.90). This point reminds us of the life-fulfilling nature of religion, which today can sometimes seem to have been discarded in the name of individualism. Even without necessarily believing in one particular religion, religion can offer deep emotions and values that can appear difficult to find in secular society. At the very least, Augustine's contrast between liberal education and religion should be considered today by emphasizing the importance of recalling these forgotten virtues that religion can provide.

## 6. Conclusions

This paper examines the role of religion in connection to rational knowledge, which is often taken for granted in today's educational purposes. The limitations of liberal education that are highlighted by Augustine remain relevant to the contemporary educational situation today: Concentrating on rationality alone cannot represent all aspects of knowledge, nor can it provide satisfactory answers to the problem of value. Despite the elevated standard of living that rational knowledge has contributed to, there persists a desire to comprehend the meanings of life and the self. In this context, religion can reclaim its significant position in discussing the foundations and purposes of our lives.

Moreover, from a philosophical perspective, the current paper serves as a bridge between Augustine's early and later thoughts. Despite his early inclinations toward liberal arts and his personal career as a rhetoric teacher, his cynical attitude to liberal education in later works is often considered a radical shift (Topping 2012, pp. 158–66), with minor exceptions (e.g., Harrison 2006). By considering dual images of Augustine as an educator, both as a rhetoric teacher and as a bishop, we can adopt a different approach to this gap.

Although whether the explanations offered by religion are true or false remains an open question, it is at least clear that we do have questions that reason cannot answer, and that yearn to be solved somehow. For one interested in these matters, the other dimension that religion illuminates cannot be taken lightly. Unlike the liberal arts, which emphasize the exploration of the external order of things, the Christian religion illuminates the inner self and strives to induce reflection on what is important to us. It also reminds us that learning is not just a process of acquiring knowledge, but a process of connecting knowledge to the inner self. In short, religion allows us to reexamine our lives by reflecting on important things beyond the reach of reason, and it gives new meaning to the knowledge that is already acquired through reason. In this respect, the religious aspect in education, which has arguably been neglected in the modern context, should still be taken seriously, as it allows for more comprehensive reflection on the full meaning of being, as well as that of education.

**Author Contributions:** Conceptualization, J.-I.L.; writing—original draft preparation, J.-I.L. and J.K.; writing—review and editing, J.K. All authors have read and agreed to the published version of the manuscript.

**Funding:** This research received no external funding.

**Institutional Review Board Statement:** Not applicable.

**Informed Consent Statement:** Not applicable.

**Data Availability Statement:** No new data were created or analyzed in this study. Data sharing is not applicable to this article.

**Conflicts of Interest:** The authors declare no conflicts of interest.

## Notes

1  In this paper, all references to Augustine's texts are provided with abbreviations and page numbers. For example, Book 1, Chapter 1, Section 1 in his *On Order* is denoted as '*ord.* 1.1.1'. The specific abbreviations in this paper include *On Order* (*de ordine → ord.*), *On True Religion* (*de vera religione → ver. rel.*), *Confessions* (*Confessiones → conf.*), *On Christian doctrine* (*De doctrina Christiana → doctr.chr.*), and *Retractations* (*Retracationes → retra.*). (St. Augustine 1959, 1968, 1992, 1997, 2020).

2  Indeed, the primary targets of Augustine's criticism are the ancient Greek philosophers Socrates and Plato (*ver. rel.* 2.2–3.3).

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
