# Peer review of "From Rational Inquiry to Sacred Insight: The Role of Religion in Augustine’s Views on Liberal Education"

_religions, doi:10.3390/rel15010122_

Round 1
Reviewer 1 Report
Comments and Suggestions for Authors
This article is concise and distinctly descriptive. It is articulated with clarity and is easily comprehensible. Its structure is transparent and logical. The primary shortcoming of the article is its purely descriptive intent as set forth by the authors. There is a lack of critical analysis of Augustine’s perspective, as well as an absence of in-depth contextualisation within the intellectual milieu of the declining Roman Empire, where Augustine’s philosophy was shaped. Nonetheless, despite these conceptual constraints, the text possesses cognitive merit and is suitable for publication.
Specific comments:
In lines 23-24, the authors write: ‘Augustine’s religious ideas have had great influence on philosophy (Spencer, 1939; Olmsted, 1989), and on education’. The mention of Augustine’s impact on philosophy, being widely recognised, is redundant. Conversely, his influence on education is less known and warrants further elaboration and clarification.
Line 53: The explanation that square brackets denote inserted, non-original text is unnecessary, as this is a standard convention.
Lines 89-90: ‘While Greek intellectuals argued that by raising the world of knowledge to the highest level, humans can have a rational mind and at the same time achieve salvation’. This statement regarding Greek intellectuals and their views on knowledge, rationality, and salvation is contentious. The authors fail to specify which Greek thinkers held these beliefs—certainly not all of them, but rather the Platonists. Moreover, ascribing the concept of salvation to Greek philosophy appears to be a misinterpretation.
Author Response
This article is concise and distinctly descriptive. It is articulated with clarity and is easily comprehensible. Its structure is transparent and logical. The primary shortcoming of the article is its purely descriptive intent as set forth by the authors. There is a lack of critical analysis of Augustine’s perspective, as well as an absence of in-depth contextualisation within the intellectual milieu of the declining Roman Empire, where Augustine’s philosophy was shaped. Nonetheless, despite these conceptual constraints, the text possesses cognitive merit and is suitable for publication.
--> We revised section 2 (Liberal Education in the Roman Era) to describe the contextual background of intellectual change and Augustine’s early thoughts on liberal education in the Roman Era.
Specific comments:
In lines 23-24, the authors write: ‘Augustine’s religious ideas have had great influence on philosophy (Spencer, 1939; Olmsted, 1989), and on education’. The mention of Augustine’s impact on philosophy, being widely recognised, is redundant. Conversely, his influence on education is less known and warrants further elaboration and clarification.
--> We revised the introduction while emphasizing liberal education more and also added some references (Line 28-37)
Line 53: The explanation that square brackets denote inserted, non-original text is unnecessary, as this is a standard convention.
--> We removed it (Line 60)
Lines 89-90: ‘While Greek intellectuals argued that by raising the world of knowledge to the highest level, humans can have a rational mind and at the same time achieve salvation’. This statement regarding Greek intellectuals and their views on knowledge, rationality, and salvation is contentious. The authors fail to specify which Greek thinkers held these beliefs—certainly not all of them, but rather the Platonists. Moreover, ascribing the concept of salvation to Greek philosophy appears to be a misinterpretation.
--> We revised the part as follows:
While some Greek intellectuals such as Plato argued that humans could attain the highest knowledge and ascent(anabasis) through the pursuit of a rational mind, acquired by inference and contemplation, Christianity instead maintained that the highest knowledge could only be achieved through the religious pursuit of God. (Line 105-109)
Reviewer 2 Report
Comments and Suggestions for Authors
This was a very frustrating essay to read. On the one hand, the author has a clear thesis and is careful in the beginning of the essay to lay out the structure of his/her argument. And the essay itself is well-organized into component parts that are clearly intended to build on each other in pursuit of its thesis (although I would add that Section 2 is unnecessary--the author should move directly into the discussion of On True Religion). Although I don't believe the thesis is groundbreaking, it is well articulated and I do leave the introductory part of the essay with an understanding what the author will argue, and how.
Then why is this essay so frustrating? The devil is in the sentence-level details. There were too many sentences I had to reread in order to understand them. Sometimes, this was a matter of odd or poor punctuation. The author has a tendency to drop in comma before "and" and "but" when those words are not operating as co-ordinating conjuctions. This often makes it difficult to parse the author's sentences, obfuscating their meaning. If this happened only on occasion, it would not be so bad. But it happens throughout the essay and repeatedly halts the reader's appreciation of the author's ideas.
But added to this are the too many times when the sentence structure frustrates understanding. On page two, for instance, the author writes: "While Greek intellectuals argued that by raising the world of knowledge to the highest level, humans can have a rational mind and at the same time achieve salvation, Christianity believed that this could only be achieve through the religious pursuit of God, because the differences between them were clear." That's tough to follow. Here's another example from page 6: "Human beings are superior to other creatures. in that they search for the meaning of existence, and are special, in that by finding purpose or meaning, they can recover themselves anew, and live their lives anew." Again, the punctation and sentence structure bury the meaning of this sentence. Perhaps other reviewers had similar difficulties.
In addition to these problems in meaning, I also had difficulty with how the author presented Section 4.3. It appears he has returned to a discussion of ideas found in On Order, but the author does not make it clear when he/she is turning back to the Augustine of On True Religion. The first paragraph of this section, then, is jarring. Also, when the author states in paragraph three that "Another limitation of knowing through reason," I am left asking, "What was the first limitation?" I had to go back and re-read the first two paragraphs to grasp that first limitation. The problems in the beginning of this section do not completely negate the point the author is making in 4.3 about the interiority of the divine order and knowledge, but it makes that point more difficult to understand.
All this to say, there is much to like about this essay, and its general point is proven, but I am wary of accepting essays that I require the reader to puzzle too often over sentences. Yes, such essays can be revised for greater clarity, but until this one is I think it is not suitable for publication.
Comments on the Quality of English Language
See my comments above.
Author Response
This was a very frustrating essay to read. On the one hand, the author has a clear thesis and is careful in the beginning of the essay to lay out the structure of his/her argument. And the essay itself is well-organized into component parts that are clearly intended to build on each other in pursuit of its thesis (although I would add that Section 2 is unnecessary--the author should move directly into the discussion of On True Religion). Although I don't believe the thesis is groundbreaking, it is well articulated and I do leave the introductory part of the essay with an understanding what the author will argue, and how.
Then why is this essay so frustrating? The devil is in the sentence-level details. There were too many sentences I had to reread in order to understand them. Sometimes, this was a matter of odd or poor punctuation. The author has a tendency to drop in comma before "and" and "but" when those words are not operating as co-ordinating conjuctions. This often makes it difficult to parse the author's sentences, obfuscating their meaning. If this happened only on occasion, it would not be so bad. But it happens throughout the essay and repeatedly halts the reader's appreciation of the author's ideas.
But added to this are the too many times when the sentence structure frustrates understanding. On page two, for instance, the author writes: "While Greek intellectuals argued that by raising the world of knowledge to the highest level, humans can have a rational mind and at the same time achieve salvation, Christianity believed that this could only be achieve through the religious pursuit of God, because the differences between them were clear." That's tough to follow. Here's another example from page 6: "Human beings are superior to other creatures. in that they search for the meaning of existence, and are special, in that by finding purpose or meaning, they can recover themselves anew, and live their lives anew." Again, the punctuation and sentence structure bury the meaning of this sentence. Perhaps other reviewers had similar difficulties.
--> First of all, we would like to thank the reviewer for their very detailed comments. Based on the reviewer’s comments, we have made substantial revisions throughout the entire paper.
Regarding Section 2, we removed some parts and substantially revised others. Although Reviewer 2 mentioned that this section was unnecessary, the other reviewer desired the addition of “in-depth contextualisation within the intellectual milieu of the declining Roman Empire, where Augustine’s philosophy was shaped”. Therefore, we changed the sub-title and attempted to specifically describe the background to Augustine’s changing views on liberal education. (Line 93 - )
In addition to these problems in meaning, I also had difficulty with how the author presented Section 4.3. It appears he has returned to a discussion of ideas found in On Order, but the author does not make it clear when he/she is turning back to the Augustine of On True Religion. The first paragraph of this section, then, is jarring. Also, when the author states in paragraph three that "Another limitation of knowing through reason," I am left asking, "What was the first limitation?" I had to go back and re-read the first two paragraphs to grasp that first limitation. The problems in the beginning of this section do not completely negate the point the author is making in 4.3 about the interiority of the divine order and knowledge, but it makes that point more difficult to understand.
--> We removed the first three paragraphs in section 4.3 (Line 326) and directly discussed liberal education in On True Religion.
Reviewer 3 Report
Comments and Suggestions for Authors
I like the article - it is simple and clear. There is one aspect, however, which I would like to suggest: the author should detail his scientific methodology. I am aware that he said how he used the sources, but he focused exclusively on primary sources. Personally, I think it would be best for the academic community - as well as the article itself - of a small section (a paragraph) detailing the scientific methodology used by the author (including his use of some key secondary sources) were included in the introduction.
Author Response
I like the article - it is simple and clear. There is one aspect, however, which I would like to suggest: the author should detail his scientific methodology. I am aware that he said how he used the sources, but he focused exclusively on primary sources. Personally, I think it would be best for the academic community - as well as the article itself - of a small section (a paragraph) detailing the scientific methodology used by the author (including his use of some key secondary sources) were included in the introduction.
--> Thank you for your kind comments. As you suggested, we added some initial statements to better guide this section and summarized the arguments made in each section. Please see the revised manuscript.
Round 2
Reviewer 2 Report
Comments and Suggestions for Authors
I say we accept this paper